# Noninvasive Diagnosis of Hepatocellular Carcinoma on Sonazoid-Enhanced US: Value of the Kupffer Phase

**DOI:** 10.3390/diagnostics12010141

**Published:** 2022-01-07

**Authors:** Hiroshi Takahashi, Katsutoshi Sugimoto, Naohisa Kamiyama, Kentaro Sakamaki, Tatsuya Kakegawa, Takuya Wada, Yusuke Tomita, Masakazu Abe, Yu Yoshimasu, Hirohito Takeuchi, Takao Itoi

**Affiliations:** 1Department of Gastroenterology and Hepatology, Tokyo Medical University, Tokyo 160-0023, Japan; h.takahashi627@gmail.com (H.T.); azusaktk36@gmail.com (T.K.); wwwtk0820@yahoo.co.jp (T.W.); toyutoyu6312@gmail.com (Y.T.); abechdesu@gmail.com (M.A.); yoshibo.you@gmail.com (Y.Y.); hirohito@yf6.so-net.ne.jp (H.T.); itoi@tokyo-med.ac.jp (T.I.); 2Ultrasound General Imaging, GE Healthcare, Hino-shi 191-0065, Japan; naohisa.kamiyama@ge.com; 3Center for Data Science, Yokohama City University, Yokohama 236-0027, Japan; kentaro.sakamaki@gmail.com

**Keywords:** ultrasound, contrast media, Sonazoid, hepatocellular carcinoma, CEUS, LI-RADS

## Abstract

The aim of this study was to compare the diagnostic performance of Contrast-Enhanced US Liver Imaging Reporting and Data System (CEUS LI-RADS) version 2017, which includes portal- and late-phase washout as a major imaging feature, with that of modified CEUS LI-RADS, which includes Kupffer-phase findings as a major imaging feature. Participants at risk of hepatocellular carcinoma (HCC) with treatment-naïve hepatic lesions (≥1 cm) were recruited and underwent Sonazoid-enhanced US. Arterial phase hyperenhancement (APHE), washout time, and echogenicity in the Kupffer phase were evaluated using both criteria. The diagnostic performance of both criteria was analyzed using the McNemar test. The evaluation was performed on 102 participants with 102 lesions (HCCs (*n* = 52), non-HCC malignancies (*n* = 36), and benign (*n* = 14)). Among 52 HCCs, non-rim APHE was observed in 92.3% (48 of 52). By 5 min, 73.1% (38 of 52) of HCCs showed mild washout, while by 10 min or in the Kupffer phase, 90.4% (47 of 52) of HCCs showed hypoenhancement. The sensitivity (67.3%; 35 of 52; 95% CI: 52.9%, 79.7%) of modified CEUS LI-RADS criteria was higher than that of CEUS LI-RADS criteria (51.9%; 27 of 52; 95% CI: 37.6%, 66.0%) (*p* = 0.0047). In conclusion, non-rim APHE with hypoenhancement in the Kupffer phase on Sonazoid-enhanced US is a feasible criterion for diagnosing HCC.

## 1. Introduction

Hepatocellular carcinoma (HCC) is the only type of cancer diagnosed noninvasively in high-risk patients on the basis of typical imaging features of CT, MRI, or contrast-enhanced ultrasound (CEUS) [1,2]. Among imaging methods, CEUS has unique advantages over CT and MRI, since it offers pure vascular images, real-time dynamic images, and excellent safety for patients with impaired renal function or allergies to iodine or gadolinium [3]. 

Recently, to improve the diagnostic accuracy for HCC and to facilitate communication among radiologists and between radiologists and other physicians, the American College of Radiology developed the Contrast-Enhanced Ultrasound Liver Imaging Reporting and Data System (CEUS LI-RADS) as a standardized reporting system for liver nodules in patients at risk for HCC [4]. 

The US contrast agents currently available are categorized into two types: pure blood pool contrast agents, such as Lumason (Bracco Diagnostics, Monroe Township, NJ, USA) and Definity (Lantheus Medical Imaging, Billerica, MA, USA), and combined blood pool and Kupffer cell contrast agents, such as Sonazoid (GE Healthcare, Oslo, Norway). 

Unfortunately, the current version of CEUS LI-RADS (version 2017) is applicable only to the pure blood pool contrast agents but not to the combined blood pool and Kupffer cell agents. This is because pure blood pool agents provide effective arterial phase hyperenhancement (APHE) and ensure pure contrast agent washout from malignant nodules. 

However, some studies have shown that in Sonazoid-enhanced US examinations, the typical dynamic enhancement patterns of HCC are hyperenhancement in the arterial phase, followed by iso-/hypoenhancement in the portal phase and defects in the Kupffer phase (post-vascular phase), the same as for blood pool agents [5]. Thus, the diagnostic accuracy of Sonazoid for HCC is comparable to that of pure blood agents.

To overcome this inconvenience, a modified CEUS LI-RADS that is also applicable to Sonazoid was developed [6]. The main difference between CEUS LI-RADS (2017 version) and the modified CEUS LI-RADS is that the former includes portal- and late-phase washout as a major imaging feature, while the latter includes Kupffer-phase findings as a major imaging feature.

Although the Kupffer phase of Sonazoid-enhanced US has been reported to facilitate the detection of small HCCs [7] and to help differentiate HCCs from hypervascular pseudolesions [7], there are some concerns about determining whether the hypoenhancement in Kupffer-phase findings can be used as an alternative to “washout”, which is warranted for the application of Sonazoid-enhanced US for the noninvasive diagnosis of HCC.

Therefore, the purpose of our study was to compare the diagnostic performance of Sonazoid-enhanced US in the noninvasive diagnosis of HCC based on the CEUS LI-RADS (2017 version) and that based on the modified CEUS LI-RADS (i.e., Kupffer-phase findings are used as an alternative to “washout”).

## 2. Materials and Methods

This study was reviewed and approved by the Tokyo Medical University ethics review board, and written informed consent was obtained from all participants. The diagnostic US scanner used (LOGIQ E10; GE Healthcare, Wauwatosa, WI, USA) was provided by the manufacturer. Only authors with no conflicts of interest had full control of the inclusion of data or information.

### 2.1. Patients

Between June 2020 and July 2021, patients who had a treatment-naïve hepatic lesion (≥1 cm) were consecutively recruited at Tokyo Medical University Hospital. The inclusion criteria were as follows: (a) age 20 years or older; (b) at risk for HCC according to American Association for the Study of Liver Diseases guidelines, which include cirrhosis of any cause and/or chronic hepatitis B [1]; (c) at least one treatment-naïve hepatic lesion (≥1 cm); (d) all nodules were visible at baseline US. When multiple eligible lesions were detected, one representative lesion per patient was analyzed.

### 2.2. US Examination

Conventional grayscale and CEUS examinations were performed by one of two hepatologists (K.S and H.T with 15 and 5 years of experience with abdominal US, respectively) using a US system (LOGIQ E10, GE Healthcare) equipped with a 3.5 MHz convex transducer (C1-6-D). The imaging mode for CEUS was the amplitude modulation method with a low mechanical index (MI) of 0.16–0.2 and a dynamic range of 63 dB. The Sonazoid contrast agent was injected as a 0.5 mL bolus into an antecubital vein via a 21-gauge peripheral intravenous cannula, followed by a 10 mL saline flush. A timer was started at the time of contrast agent injection. A targeted lesion was recorded continuously as a cine clip for 60 s after injection. During this period, the patient was instructed to breathe gently. After that, the same targeted lesion was recorded at one-minute intervals as a five-second cine clip with breath hold. This occurred between the 2-min mark and the 10-min mark. The sequence that was followed in the CEUS protocol is shown in Figure 1.

### 2.3. Image Analysis

One hepatologist with more than 15 years of experience in liver CEUS who was blinded to reference standard results and other imaging test results reviewed CEUS of liver nodules and assigned a category according to CEUS LI-RADS (v2017) [4] and modified CEUS LI-RADS [6]. Briefly, the main difference between CEUS LI-RADS (v2017) and the modified CEUS LI-RADS is that the former includes portal-phase and late-phase washout as major imaging features, while the latter includes Kupffer-phase findings as a major imaging feature. The other criteria in the modified CEUS LI-RADS are almost the same as those in CEUS LI-RADS (v2017). The following diagnostic features were used to characterize each nodule based on CEUS LI-RADS and modified CEUS LI-LADS: nodule size; arterial phase enhancement and its pattern; the presence, timing, and degree of washout until 5 or 10 min after injection; and mosaic and nodule-in-nodule architecture. Note that the predetermined time limit for washout was 5 min in CEUS LI-RADS (v2017) and was 10 min or the Kupffer phase in modified CEUS LI-RADS.

### 2.4. Reference Standard

Of the target lesions, 78.4% (80 of 102) were diagnosed histopathologically (surgery, *n* = 2; biopsy, *n* = 78). Information on hepatic tumor pathology and immunohistochemistry were routinely described in pathologic reports at our institution. A total of 14.7% (15 of 102) of target lesions were noninvasively diagnosed as HCC because they were categorized as LR-5 according to CT/MRI LI-RADS v2018 [1]. Five (4.9%) cases of targeted lesions showed peripheral globular and centripetal enhancement. Thus, these were regarded as hemangiomas without pathologic confirmation. Two (2.0%) cases of the targeted lesions were first presumed to be LR-3. These two LR-3 lesions showed high signal intensity on T2-weighted MRI during the follow-up period and did not show threshold growth for more than 6 months. Thus, eventually, these were also regarded as hemangiomas without pathologic confirmation. In addition, the diagnosis of liver cirrhosis was made by using ultrasound elastography. When the liver stiffness value was more than 13 kPa, we regarded the condition as liver cirrhosis [8].

### 2.5. Statistical Analysis

The per-lesion sensitivity, specificity, and accuracy of both CEUS LI-RADS criteria and modified CEUS LI-RADS criteria were compared by using the McNemar test. Correlation between Kupffer-phase enhancement and grayscale echogenicity on HCC, correlation between Kupffer-phase enhancement and HCC differentiation, and correlation between early washout and HCC differentiation were analyzed by using a χ^2^ test. All statistical analyses were performed by using SAS software, version 9.4 (SAS Institute). A one-tailed *p* value less than 0.05 was considered to indicate a significant difference.

## 3. Results

### 3.1. Patients and Liver Nodule Characteristics

A total of 113 participants met the inclusion criteria. Among them, 11 were excluded because of poor-quality images for analysis (*n* = 5) and incomplete diagnosis by CT/MRI (*n* = 6). Accordingly, 102 patients with 102 lesions (male: 64, female: 48; median age and interquartile range, 71 years and 63–78 years) were finally included in this study (Figure 2). The clinical and pathologic characteristics of the participants and the target lesions are described in Table 1. The median size of the observed lesions was 25.5 mm (interquartile range, 16.8–44.3 mm). Of these lesions, 51.0% (52 of 102) were confirmed as HCCs, 35.3% (36 of 102) were non-HCC malignancies, and 13.7% (14 of 102) were benign lesions. Non-HCC malignancies included 26 metastases and 10 intrahepatic cholangiocarcinomas. The clinical and pathologic characteristics of the participants with liver metastases and the target lesions are described in Table 2. Benign lesions included seven hemangiomas, five focal nodular hyperplasias, and one angiomyolipoma. The most common cause of liver disease was alcohol abuse (40.2% [41 of 102]), and 79.4% (81 of 102) of participants had cirrhosis.

### 3.2. Major Imaging Features of Contrast-Enhanced US

#### 3.2.1. Arterial Phase

The arterial phase characteristics of 102 examined liver nodules are listed in Table 3. Non-rim arterial hyperenhancement (APHE) was observed in 92.3% (48 of 52) of HCCs and 41.7% (15 of 36) of non-HCC malignancies. Rim APHE was observed in 19.4% (7 of 36) of non-HCC malignancies. Dysplastic nodules (*n* = 1) had isoenhancement in the arterial phase. Peripheral globular and centripetal enhancement was observed in 71.4% (five of seven) of hemangioma, and all focal nodular hyperplasia (*n* = 5) and angiomyolipoma (*n* = 1) showed non-rim APHE.

#### 3.2.2. Washout

The washout characteristics of 102 examined liver nodules are listed in Table 4. Early washout was observed in 17.3% (9 of 52) of HCCs. By 5 min, 73.1% (38 of 52) of HCCs showed mild washout, while by 10 min or in the Kupffer phase, 90.4% (47 of 52) of HCCs showed hypoenhancement. Five HCCs did not show washout until 10 min. Early washout was observed in 91.7% (33 of 36) of non-HCC malignancies, and by four minutes, 100% (36 of 36) of non-HCC malignancies showed washout. None of the benign lesions had early washout. By both 5 and 10 min, 14.3% (three of 14) benign lesions showed mild washout.

### 3.3. Diagnostic Performance of Hepatocellular Carcinoma Diagnosis Based on Both CEUS LI-RADS and Modified CEUS LI-RADS Using Qualitative Washout Results

The sensitivity, specificity, and accuracy for HCC diagnosis based on CEUS LI-RADS (2017 version) and modified CEUS LI-RADS are summarized in Table 5. The sensitivity (67.3%; 35 of 52; 95% CI: 52.9%, 79.7%) of modified CEUS LI-RADS criteria was higher than that of CEUS LI-RADS criteria (51.9%; 27 of 52; 95% CI: 37.6%, 66.0%) (*p* = 0.0047). The specificity was the same (92.0%; 46 of 50; 95% CI: 80.8%, 97.8%) for both criteria. The accuracy (79.4%; 81 of 102; 95% CI: 70.3%, 86.8%) of the modified CEUS LI-RADS criteria was higher than that of the CEUS LI-RADS criteria (71.6%; 73 of 102; 95% CI: 61.8%, 80.1%) (*p* = 0.0047).

## 4. Discussion

Our results show that Sonazoid-enhanced US with the modified CEUS LI-RADS criteria, where Kupffer-phase findings were used as an alternative to “washout”, provided higher sensitivity (67.3%) and accuracy (0.794) for noninvasive hepatocellular carcinoma (HCC) diagnosis compared with CEUS LI-RADS (2017 version), in which the predetermined time limit for washout was 5 min (sensitivity, 51.9%; *p* = 0.0047; accuracy, 0.794; *p* = 0.0047). Moreover, higher sensitivity and accuracy were achieved without losing specificity.

These results were due to the difference in when we should determine washout, without differences in wash-in patterns. Our results show that the longer we waited, the more HCCs exhibited washout. After 5 min, 38 out of 52 HCCs (73.1%) showed washout, while after 10 min, the so-called Kupffer phase, 47 out of 52 HCCs (90.4%) showed washout (Figure 3). Although there were some concerns about determining whether the hypoenhancement in Kupffer-phase findings can be used as an alternative to “washout”, which is warranted for the application of Sonazoid-enhanced US for the noninvasive diagnosis of HCC, our results suggest that washout should be assessed in the Kupffer phase rather than before for a more specific diagnosis of HCCs on Sonazoid-enhanced US. Thus, Kupffer-phase findings can be used as an alternative to “washout”.

The same question was raised for gadolinium-ethoxybenzyl-diethylenetriamine pentaacetic acid magnetic resonance imaging (EOB-MRI) about whether the hypointensity observed in dynamic phases and/or the hepatobiliary phase (HBP) can be used as an alternative to “washout” for the noninvasive diagnosis of HCC. In this regard, Joo et al. [9] explored the question and demonstrated that the criterion of “hypointensity on the PVP” of EOB-MRI resulted in the high specificity of 97.9%, while the criteria of “hypointensity on the PVP and/or TP, and/or HBP” resulted in lower specificities of 86.3% and 48.4%, respectively. Thus, they concluded that washout should be assessed in the PVP alone rather than combined with the TP or HBP for a more specific diagnosis of HCC on EOB-MRI. It should be noted that the result of EOB-MRI was contrary to that of Sonazoid-enhanced US.

Of all HCCs, there were five HCCs (9.6%) that showed isoenhancement in the Kupffer phase, leading to misclassification as LR-4 (Figure 4). We investigated the correlation between Kupffer-phase enhancement and grayscale features and found that all HCC nodules that showed isoenhancement in the Kupffer phase were isoechoic in the grayscale US image (Table 6). Thus, hyperechoic lesions may not appear as a defect in the Kupffer phase in the medium-MI (i.e., MI values from 0.1 to 0.2) contrast imaging mode, compared with background signals. To overcome this and enhance the diagnostic accuracy of HCC, high-MI contrast imaging may be useful because it is more sensitive than low-MI contrast imaging for detecting whether there were microbubbles or not in HCC nodules [10,11] (Figure 5, Appendix A).

In addition, of 37 HCCs confirmed by pathologic analysis, there were 4 HCCs (10.8%) that showed isoenhancement in the Kupffer phase, leading to misclassification as LR-4. The correlation between Kupffer-phase enhancement and hepatocellular carcinoma differentiation is in Table 7. Among these four HCCs, three were well-differentiated HCCs. In contrast, almost all moderately differentiated (22 of 23 nodules: 95.7%) and all poorly differentiated HCCs (5 of 5 nodules: 100%) showed hypoenhancement in the Kupffer phase. This is because Kupffer cells remain inside dysplastic nodules and early HCC lesions, which contain blood spaces that are more like normal sinusoids [12]. Thus, it should be noted that well-differentiated HCCs were also less likely to show hypoenhancement in the Kupffer phase.

Among 52 HCCs in the present study, washout starting within 60 s or early washout was observed in 9 nodules (17.3%). In addition, of 37 HCCs confirmed by pathologic analysis, there were 8 HCCs (21.6%) that showed early washout, leading to misclassification as LR-M. The correlation between early washout and hepatocellular carcinoma differentiation is shown in Table 8. Among the eight HCCs that showed early washout, five were poorly differentiated HCCs. In contrast, almost all moderately differentiated (20 of 23 nodules: 87.0%) and all well-differentiated HCCs (9 of 9 nodules: 100%) did not show early washout. These findings are in line with previous studies [6,13].

Although the early washout feature for poorly differentiated HCCs decreased the diagnostic accuracy of HCC, this finding is actually rather advantageous because poorly differentiated HCCs have higher malignant potential than other types of HCCs and are therefore more similar in nature to metastases [14].

In this study, we did not make a direct comparison with blood pool agents, which are now applicable to CEUS LI-RADS (version 2017). According to a prior report [15], perfluorobutane-enhanced US (i.e., Kupffer cell agent) was superior to sulfur hexafluoride-enhanced US (i.e., blood pool agent) in the diagnostic performance of HCC. This suggests that Sonazoid has much more potential for application in daily clinical practice and can be used on a regular basis in the future.

The present study had several limitations. First, we used a single-center population. Thus, multicenter and multinational studies are warranted to validate our study results. Second, the relatively small number of benign lesions (14 of 102: 13.7%) hindered the generalizability of our results. Moreover, only half of the benign lesions were examined histopathologically. However, in the study, 37 of 52 (71.2%) HCCs and all non-HCC malignancies were diagnosed by histopathologic examination, which is a major strength of the study. Finally, although Kupffer-phase findings were useful for diagnosing HCC, the 10 min observation period is a disadvantage of Sonazoid-enhanced US due to its time-consuming nature.

In conclusion, Sonazoid-enhanced US is feasible for diagnosing HCC in individuals with high risk based on non-rim arterial phase hyperenhancement and subsequent Kupffer-phase hypoenhancement. The modified CEUS LI-RADS (i.e., Kupffer-phase findings are used as an alternative to “washout”) was more useful than CEUS LI-RADS (2017 version) in diagnosing HCC. Therefore, washout should be determined in the Kupffer phase or post-vascular phase to maintain the high diagnostic accuracy of HCC.

## Figures and Tables

**Figure 1 diagnostics-12-00141-f001:**
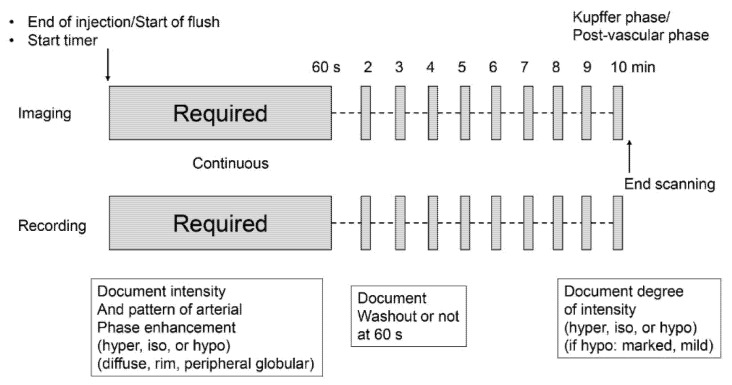
Sonazoid Contrast-Enhanced Ultrasound protocol.

**Figure 2 diagnostics-12-00141-f002:**
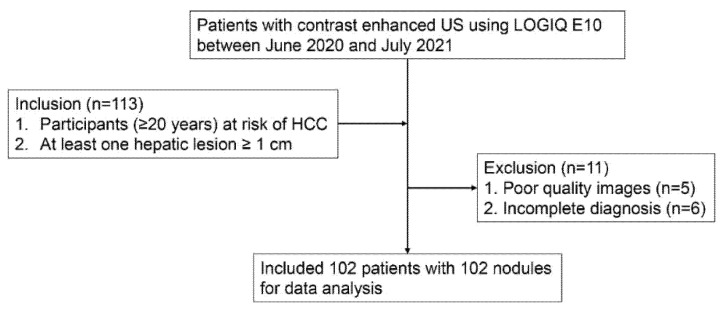
Flow diagram of study.

**Figure 3 diagnostics-12-00141-f003:**
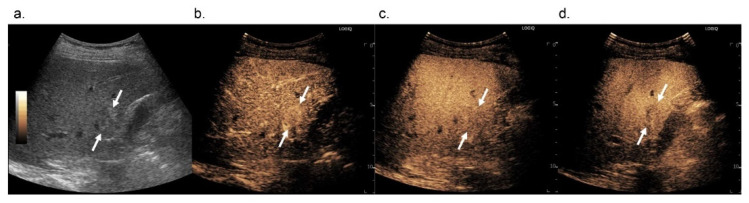
Images in 74-year-old man with pathologically confirmed hepatocellular carcinoma in segment 5 of the liver. On B-mode US, a 1.7-cm hyperechoic nodule is observed (**a**, arrows). On Sonazoid-enhanced US, the nodule shows arterial phase hyperenhancement (**b**, arrows) and does not show washout until 5 min after contrast media injection (**c**, arrows). The nodule shows hypoenhancement in the Kupffer phase (**d**, arrows).

**Figure 4 diagnostics-12-00141-f004:**
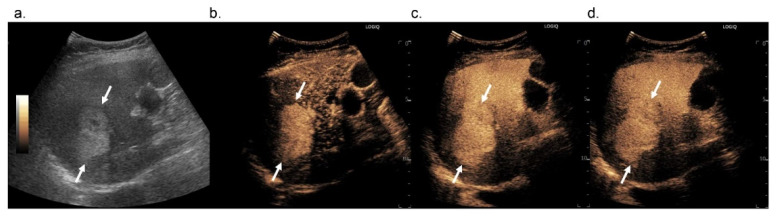
Images in 72-year-old woman with pathologically confirmed hepatocellular carcinoma in segment 8 of the liver. In B-mode US, a 5.5 cm hyperechoic nodule is observed (**a**, arrows). In Sonazoid-enhanced US, the nodule shows arterial phase hyperenhancement (**b**, arrows) and does not show washout until 5 min after contrast media injection (**c**, arrows). The nodule shows isoenhancement in the Kupffer phase (**d**, arrows).

**Figure 5 diagnostics-12-00141-f005:**
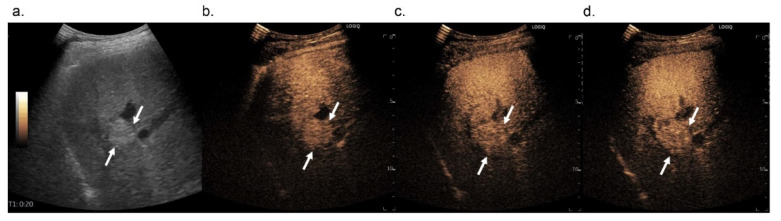
Images in 81-year-old man with pathologically confirmed hepatocellular carcinoma in segment 8 of the liver. In B-mode US, a 2.8 cm hyperechoic nodule is observed (**a**, arrows). In Sonazoid-enhanced US, the nodule shows arterial phase hyperenhancement (**b**, arrows) and does not show washout until 5 min after contrast media injection (**c**, arrows). The nodule shows isoenhancement in the Kupffer phase (**d**, arrows).

**Table 1 diagnostics-12-00141-t001:** Clinical and histopathologic data (*n* = 102).

Characteristic	Result
Median age (year) *	71 (63–78)
Sex	
Male	64 (62.7)
Female	38 (37.3)
Median nodule size (mm) *	25.5 (16.8–44.3)
Liver disease etiology	
Alcohol	41 (40.2)
HBV	30 (29.4)
HCV	18 (17.6)
NASH	13 (12.7)
Presence of cirrhosis	81 (79.4)
Histopathologic analysis	80 (78.4)
HCC	37 (36.3)
Well-differentiated	9 (24.3)
Moderately differentiated	23 (62.2)
Poorly differentiated	5 (13.5)
Metastasis	26 (25.5)
ICC	10 (9.8)
FNH	5 (4.9)
Dysplastic nodule	1 (1.0)
AML	1 (1.0)
No histopathologic analysis	22 (21.6)
Contrast-enhanced CT or MRI	
HCC	15 (14.7)
Hemangioma	7 (6.9)

Note: Data are numbers of patient observations, with percentages in parentheses. * Data in parentheses are interquartile range. HBV = hepatitis B virus, HCV = hepatitis C virus, NASH = nonalcoholic steatohepatitis, HCC = hepatocellular carcinoma, ICC = intrahepatic cholangiocarcinoma, FNH = focal nodular hyperplasia, AML = angiomyolipoma, and y = year.

**Table 2 diagnostics-12-00141-t002:** Clinical and histopathologic data of liver metastases (*n* = 26).

Characteristic	Result
Median age (year) *	69 (61–74)
Sex	
Male	15 (57.7)
Female	11 (42.3)
Median nodule size (mm) *	24.0 (14.8–65.8)
Liver disease etiology	
Alcohol	17 (65.4)
HBV	8 (30.8)
NASH	1 (3.8)
HCV	0 (0)
Presence of cirrhosis	18 (69.2)
Origin or histopathologic types of the metastases	
Breast	5 (19.5)
Pancreatic neuroendocrine tumor	4 (15.4)
Stomach	4 (15.4)
Diffuse large B-cell lymphoma	3 (11.5)
Malignant melanoma	2 (7.7)
Pancreas	2 (7.7)
Adenoid cystic carcinoma	1 (3.8)
Colon	1 (3.8)
Ovary	1 (3.8)
Kidney	1 (3.8)
Sebaceous carcinoma of the eyelid	1 (3.8)
Unknown	1 (3.8)
No histopathologic analysis	0 (0)

Note: Data are numbers of patient observations, with percentages in parentheses. * Data in parentheses are interquartile range. HBV = hepatitis B virus, HCV = hepatitis C virus, NASH = nonalcoholic steatohepatitis, and y = year.

**Table 3 diagnostics-12-00141-t003:** Arterial phase characteristics.

Characteristics of Arterial Phase	HCC (*n* = 52)	Non-HCC Malignancy			Benign Lesions			
		Metastasis (*n* = 24)	ICC (*n* = 10)	Lymphoma (*n* = 2)	Hemangioma (*n* = 7)	FNH (*n* = 5)	DN (*n* = 1)	AML (*n* = 1)
Intensity								
Hyperenhancement	48	13	2	0	3	5	0	1
Isoenhancement	4	9	4	2	0	0	1	0
Hypoenhancement	0	2	4	0	4	0	0	0
Pattern								
Diffuse	52	20	7	2	2	5	1	1
Rim	0	4	3	0	0	0	0	0
Peripheral nodular	0	0	0	0	5	0	0	0

Note: Data are numbers of findings. HCC = hepatocellular carcinoma, ICC = intrahepatic cholangiocarcinoma, FNH = focal nodular hyperplasia, DN = dysplastic nodule, and AML = angiomyolipoma.

**Table 4 diagnostics-12-00141-t004:** Washout characteristics.

		No. of Washout and Its Ratio According to Time									
Diagnosis	No. of Target Lesions	1 min or Early Washout	2 min	3 min	4 min	5 min	6 min	7 min	8 min	9 min	10 min or Kupffer Phase
Qualitative analysis											
HCCs	52	9 (17.3)	20 (38.5)	29 (55.8)	32 (61.5)	38 (73.1)	40 (76.9)	42 (80.8)	44 (84.6)	45 (86.5)	47 (90.4)
Non-HCC malignancies	36	33 (91.7)	34 (94.4)	35 (97.2)	36 (100)	36 (100)	36 (100)	36 (100)	36 (100)	36 (100)	36 (100)
Benign lesions	14	0 (0)	3 (21.4)	3 (21.4)	3 (21.4)	2 (14.3)	3 (21.4)	3 (21.4)	2 (14.3)	3 (21.4)	2 (14.3)

Note: Values are numbers, with percentages in parentheses. HCC = hepatocellular carcinoma.

**Table 5 diagnostics-12-00141-t005:** Diagnostic performance of hepatocellular carcinoma diagnosis based on both LI-RADS and modified LI-RADS criteria.

Variable	CEUS LI-RADS	Modified CEUS LI-RADS	*p* Value
Sensitivity	0.519 (0.376–0.660)	0.673 (0.529–0.797)	0.0047
Specificity	0.920 (0.808–0.978)	0.920 (0.808–0.978)	n.s.
Accuracy	0.716 (0.618–0.801)	0.794 (0.703–0.868)	0.0047

Note: Data in parentheses are 95% confidence intervals. CEUS LI-RADS = Contrast-Enhanced Ultrasound Liver Imaging Reporting and Data System, and n.s. = not significant.

**Table 6 diagnostics-12-00141-t006:** Correlation between Kupffer-phase enhancement and grayscale echogenicity on hepatocellular carcinoma.

Kupffer-Phase Enhancement	Hyperechoic	Hypoechoic	Isoechoic	Total
Hypo	16 (30.8)	28 (53.8)	3 (5.8)	47 (90.4)
Iso	5 (9.6)	0 (0)	0 (0)	5 (9.6)
Total	21 (40.4)	28 (53.8)	3 (5.8)	52 (100)

Note: Values are numbers, with percentages in parentheses: *p* = 0.0169.

**Table 7 diagnostics-12-00141-t007:** Correlation between Kupffer-phase enhancement and hepatocellular carcinoma differentiation.

Kupffer-Phase Enhancement	Well-Differentiated HCC	Moderately Differentiated HCC	Poorly Differentiated HCC	Total
Hypo	6 (16.2)	22 (59.5)	5 (13.5)	33 (89.2)
Iso	3 (8.1)	1 (2.7)	0 (0)	4 (10.8)
Total	9 (24.3)	23 (62.2)	5 (13.5)	37 (100)

Note: Values are numbers, with percentages in parentheses. HCC = hepatocellular carcinoma: *p* = 0.0421.

**Table 8 diagnostics-12-00141-t008:** Correlation between early washout and hepatocellular carcinoma differentiation.

Early Washout	Well-Differentiated HCC	Moderately Differentiated HCC	Poorly Differentiated HCC	Total
Yes	0 (0)	3 (8.1)	5 (13.5)	8 (21.6)
No	9 (24.3)	20 (54.1)	0 (0)	29 (78.4)
Total	9 (24.3)	23 (62.2)	5 (13.5)	37 (100)

Note: Values are numbers, with percentages in parentheses. HCC = hepatocellular carcinoma: *p* < 0.0001.

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
