# Peer review of "Noninvasive Diagnosis of Hepatocellular Carcinoma on Sonazoid-Enhanced US: Value of the Kupffer Phase"

_diagnostics, 2022, doi:10.3390/diagnostics12010141_

Round 1
Reviewer 1 Report
- Rows 68-81 I think should be deleted
- As described in the material and method chapter, patients with suspected CT and MRI hemangiomas were not included in the study. However, you have patients with hepatic hemangiomas included in your study. How was the final diagnosis (gold standard) made?
- Using CEUS you detected patients with liver metastases. Please detail this subgroup of patients: if they were patients with liver cirrhosis, what is the histopathological type and location of the primary tumor (it is known that the liver with cirrhosis rarely receives metastases) and what is the ultrasound pattern of these tumors: hyper / hypovascular.
- You only found one patient with a dysplastic nodule. What is the pattern of this dysplastic node? Was it a typical look? Have you experienced problems with the differential diagnosis of dysplastic nodules? If so, what were the diagnostic issues?
Author Response
Point 1: Rows 68-81 I think should be deleted
Response 1: Thank you for pointing this out. According to your suggestion, we have delted this part.
Point 2: As described in the material and method chapter, patients with suspected CT and MRI hemangiomas were not included in the study. However, you have patients with hepatic hemangiomas included in your study. How was the final diagnosis (gold standard) made?
Response 2: Thank you for pointing this out. We have revised “2.4. Reference Standerd” section accordingly (page 3-4, lines 128-136).
Point 3: Using CEUS you detected patients with liver metastases. Please detail this subgroup of patients: if they were patients with liver cirrhosis, what is the histopathological type and location of the primary tumor (it is known that the liver with cirrhosis rarely receives metastases) and what is the ultrasound pattern of these tumors: hyper / hypovascular.
Response 3:
Thank you for your important suggestions. We will provide you with the detail information on liver metastasis as a Table 1.2. Also, the diagnosis of cirrhosis was made by using ultrasound elastography. When the liver stiffness value was more than 13 kPa, we regarded as liver cirrhosis. Although we performed liver biopsy to diagnose liver nodule in this study, the diagnosis of cirrhosis was difficult to made using needle aspiration biopsy. Thus we used ultrasond elastography as reference standerd of liver cirrhosis. We have added the discription in “2.4. Reference Standard” section. As for the ultrasound pattern of these tumors has been already described in Table 2.
On table 1.1., diffuse larege B cell lymphomas should be included iin liver metastases. Thus, we have delated it and added to liver metastasis.
Table 1.2. Clinical and histopathologic data of liver metastases (n=26)
Characteristic |
Result |
Median age (y) * |
69 (61-74) |
Sex |
|
Male |
15 (57.7) |
Female |
11 (42.3) |
Median nodule size (mm) * |
24.0 (14.8-65.8) |
Liver disease etiology |
|
Alcohol |
17 (65.4) |
HBV |
8 (30.8) |
NASH |
1 (3.8) |
HCV |
0 (0 |
Presence of cirrhosis |
18 (69.2) |
Origin or histopathologic types of the metastases |
|
Breast |
5 (19.5) |
Pancreatic neuroendocrine tumor |
4 (15.4) |
Stomach |
4 (15.4) |
Diffuse large B-cell lymphoma |
3 (11.5) |
Malignant melanoma |
2 (7.7) |
Pancreas |
2 (7.7) |
Adenoid cystic carcinoma |
1 (3.8) |
Colon |
1 (3.8) |
Ovary |
1 (3.8) |
Kidny |
1 (3.8) |
Sebaceous carcinoma of the eyelid |
1 (3.8) |
Unknown |
1 (3.8) |
No histopathologic analysis |
0 (0) |
Point 4: You only found one patient with a dysplastic nodule. What is the pattern of this dysplastic node? Was it a typical look? Have you experienced problems with the differential diagnosis of dysplastic nodules? If so, what were the diagnostic issues?
Response 4:
In this casa, the nodule was 19 mm in diameter and hypo echoic apperarence on B-mode US. On CEUS, the nodule showed iso-vasucularity in AP and persisted it untill Kupffer pahse. Although it was typical apperarence of DN, tumor biopsy was nessessary for final diagnosis.
Reviewer 2 Report
The authors investigate the diagnostic performance by comparing Contrast-Enhanced US Liver Reporting and Data System (CEUS LI-RADS) version 2017 which includes portal and late phase washout as a major imaging feature and modified CEUS LI-RADS which includes Kupffer phase findings as a major imaging features.
Although the liver is the organ to which cancer cells most frequently metastasize for the majority of prevalent malignancies, in liver cirrhosis the prevalence of metastases is very low. In the present study, the percent of the metastases are very high: 23% of the all lesions. Moreover, the liver cirrhosis is present in 79% of cases. Please give some explanations and give details about the origin of the metastases, status of the liver disease (stage of fibrosis, presence of cirrhosis) and the method of final diagnosis for each case.
Author Response
The authors investigate the diagnostic performance by comparing Contrast-Enhanced US Liver Reporting and Data System (CEUS LI-RADS) version 2017 which includes portal and late phase washout as a major imaging feature and modified CEUS LI-RADS which includes Kupffer phase findings as a major imaging feature.
Point 1: Although the liver is the organ to which cancer cells most frequently metastasize for the majority of prevalent malignancies, in liver cirrhosis the prevalence of metastases is very low. In the present study, the percent of the metastases are very high: 23% of all the lesions. Moreover, the liver cirrhosis is present in 79% of cases. Please give some explanations and give details about the origin of the metastases, status of the liver disease (stage of fibrosis, presence of cirrhosis) and the method of final diagnosis for each case
Response 1:
Thank you for your important suggestions. We will provide you with the detail information on liver metastasis as a Table 1.2. Also, the diagnosis of cirrhosis was made by using ultrasound elastography. When the liver stiffness value was more than 13 kPa, we regarded as liver cirrhosis. Although we performed liver biopsy to diagnose liver nodule in this study, the diagnosis of cirrhosis was difficult to made using needle aspiration biopsy. Thus we used ultrasond elastography as reference standerd of liver cirrhosis. We have added the discription in “2.4. Reference Standard” section.
On table 1.1., diffuse larege B cell lymphomas should be included iin liver metastases. Thus, we have delated it and added to liver metastasis.
Table 1.2. Clinical and histopathologic data of liver metastases (n=26)
Characteristic |
Result |
Median age (y) * |
69 (61-74) |
Sex |
|
Male |
15 (57.7) |
Female |
11 (42.3) |
Median nodule size (mm) * |
24.0 (14.8-65.8) |
Liver disease etiology |
|
Alcohol |
17 (65.4) |
HBV |
8 (30.8) |
NASH |
1 (3.8) |
HCV |
0 (0 |
Presence of cirrhosis |
18 (69.2) |
Origin or histopathologic types of the metastases |
|
Breast |
5 (19.5) |
Pancreatic neuroendocrine tumor |
4 (15.4) |
Stomach |
4 (15.4) |
Diffuse large B-cell lymphoma |
3 (11.5) |
Malignant melanoma |
2 (7.7) |
Pancreas |
2 (7.7) |
Adenoid cystic carcinoma |
1 (3.8) |
Colon |
1 (3.8) |
Ovary |
1 (3.8) |
Kidny |
1 (3.8) |
Sebaceous carcinoma of the eyelid |
1 (3.8) |
Unknown |
1 (3.8) |
No histopathologic analysis |
0 (0) |
Reviewer 3 Report
This study entitled “Noninvasive diagnosis of hepatocellular carcinoma on Sonazoid-enhanced US: Value of the Kupffer phase” is very interesting and has important useful implications in clinical practice related to the diagnosis of HCC.
Since the end of the 90s of the last century has been used ultrasound technique with contrast medium to improve the detection of HCC (Contrast-Enhanced Ultrasound –CEUS) in patients at risk. Then after about 20 years from the introduction of CEUS, to improve the diagnostic accuracy of the examination and to make homogeneous the response in order to facilitate communication among radiologists and between radiologists and other physicians, the American College of Radiology has developed the Contrast-Enhanced Ultrasound Liver Imaging Reporting and Data System (CEUS LI-RADS) as a standardized reporting system for liver nodules in patients at risk for HCC.
The latter is based on the use of contrast medium currently most widespread in the world (Sonovue, Bracco). However, the introduction already in the early 2000s of a new contrast medium for CEUS (Sonazoid (GE Healthcare, Waukesha, WI, USA) with slightly different characteristics related to the analysis of late phases thanks to the diffusion of the medium in the Kupffer phase (post-vascular phase). For this reason it was necessary to verify whether a new CEUS LI-RADS protocol (modified CEUS LI-RADS) was more useful than the previous one.
The purpose of this paper is therefore to compare these two evaluation modes (CEUS LI-RADS standard vs modified CEUS LI-RADS) in patients where CEUS is performed with Sonazoid.
Based on the results of the paper, the authors conclude that there is evidence of a significant improvement in the sensitivity of the new method (modified CEUS LI-RADS) compared to the traditional one with no significant change in specificity and improved diagnostic accuracy from 71.6 to 79.4%.
In conclusion the paper is well conducted from a methodological point of view and does not present particular weaknesses apart from those mentioned by the authors themselves in the discussion. In particular, the results come from a single center (there is therefore no external validation); moreover, only half of the benign liver lesions had been subjected to liver biopsy.
Minor limitations of the study
- There is no sample size calculation to assess the minimum number of patients to be recruited to achieve adequate statistical power. In this way the conclusions drawn from the results obtained by the authors become more reliable. The main way to achieve adequate power is to plan an adequate sample size in the study protocol
- In materials and methods, 2.2 US Examination (page 3, lines 105-106) the authors report “A targeted lesion was recorded continuously as a cine clip for 60 minutes after injection.”. It is unclear whether a recording was actually made for that period of time or (most likely) this is a typo for 60 seconds. It needs to be clarified and corrected.
- Finally, the entire period (page 2, lines 72-80) is not clear. It refers to advice reported by another referee and by mistake reported in the paper (“The introduction should briefly place the study in a broad context and highlight why 72 it is important. It should define the purpose of the work and its significance. The current 73 state of the research field should be carefully reviewed and key publications cited. Please 74 highlight controversial and diverging hypotheses when necessary. Finally, briefly men- 75 tion the main aim of the work and highlight the principal conclusions. As far as possible, 76 please keep the introduction comprehensible to scientists outside your particular field of 77 research. References should be numbered in order of appearance and indicated by a nu- 78 meral or numerals in square brackets—e.g., [1] or [2,3], or [4–6]. See the end of the docu- 79 ment for further details on references.”).
In conclusion the paper seems to be really interesting and well conducted. It can be a starting point for further investigation and confirmation in this field while offering useful elements to be applied in clinical practice.
Author Response
This study entitled “Noninvasive diagnosis of hepatocellular carcinoma on Sonazoid-enhanced US: Value of the Kupffer phase” is very interesting and has important useful implications in clinical practice related to the diagnosis of HCC.
Since the end of the 90s of the last century has been used ultrasound technique with contrast medium to improve the detection of HCC (Contrast-Enhanced Ultrasound–CEUS) in patients at risk. Then after about 20 years from the introduction of CEUS, to improve the diagnostic accuracy of the examination and to make homogeneous the response in order to facilitate communication among radiologists and between radiologists and other physicians, the American College of Radiology has developed the Contrast-Enhanced Ultrasound Liver Imaging Reporting and Data System (CEUS LI-RADS) as a standardized reporting system for liver nodules in patients at risk for HCC.
The latter is based on the use of contrast medium currently most widespread in the world (Sonovue, Bracco). However, the introduction already in the early 2000s of a new contrast medium for CEUS (Sonazoid, GE Healthcare, Waukesha, WI, USA) with slightly different characteristics related to the analysis of late phases thanks to the diffusion of the medium in the Kupffer phase (post-vascular phase). For this reason it was necessary to verify whether a new CEUS LI-RADS protocol (modified CEUS LI-RADS) was more useful than the previous one.
The purpose of this paper is therefore to compare these two evaluation modes (CEUS LI-RADS standard vs modified CEUS LI-RADS) in patients where CEUS is performed with Sonazoid.
Based on the results of the paper, the authors conclude that there is evidence of a significant improvement in the sensitivity of the new method (modified CEUS LI-RADS) compared to the traditional one with no significant change in specificity and improved diagnostic accuracy from 71.6 to 79.4%.
In conclusion the paper is well conducted from a methodological point of view and does not present particular weaknesses apart from those mentioned by the authors themselves in the discussion. In particular, the results come from a single center (there is therefore no external validation); moreover, only half of the benign liver lesions had been subjected to liver biopsy.
Minor limitations of the study
Point 1: There is no sample size calculation to assess the minimum number of patients to be recruited to achieve adequate statistical power. In this way the conclusions drawn from the results obtained by the authors become more reliable. The main way to achieve adequate power is to plan an adequate sample size in the study protocol
Response 1: Thank you for your suggesiton. Unfortunately, we did not perform sample size estimation because this stady was an exploratory study.
Point 2: In materials and methods, 2.2 US Examination (page 3, lines 105-106) the authors report “A targeted lesion was recorded continuously as a cine clip for 60 minutes after injection.”. It is unclear whether a recording was actually made for that period of time or (most likely) this is a typo for 60 seconds. It needs to be clarified and corrected.
Response 2: Thank you for pointing it out. We have corrected it.
Point 3: Finally, the entire period (page 2, lines 72-80) is not clear. It refers to advice reported by another referee and by mistake reported in the paper (“The introduction should briefly place the study in a broad context and highlight why 72 it is important. It should define the purpose of the work and its significance. The current 73 state of the research field should be carefully reviewed and key publications cited. Please 74 highlight controversial and diverging hypotheses when necessary. Finally, briefly men- 75 tion the main aim of the work and highlight the principal conclusions. As far as possible, 76 please keep the introduction comprehensible to scientists outside your particular field of 77 research. References should be numbered in order of appearance and indicated by a nu- 78 meral or numerals in square brackets—e.g., [1] or [2,3], or [4–6]. See the end of the docu- 79 ment for further details on references.”).
Response 3: Thank you for pointing it out. It was simmple mistake. We have removed the part.
In conclusion the paper seems to be really interesting and well conducted. It can be a starting point for further investigation and confirmation in this field while offering useful elements to be applied in clinical practice.